# Canadian Adults with Moderate Intakes of Total Sugars have Greater Intakes of Fibre and Key Micronutrients: Results from the Canadian Community Health Survey 2015 Public Use Microdata File

**DOI:** 10.3390/nu12041124

**Published:** 2020-04-17

**Authors:** Ye (Flora) Wang, Laura Chiavaroli, Kaitlin Roke, Chiara DiAngelo, Sandra Marsden, John Sievenpiper

**Affiliations:** 1Nutrition Information Service, Canadian Sugar Institute, Toronto, ON M5V 3E4, Canada; 2Department of Nutritional Sciences, University of Toronto, Toronto, ON M5B 1W8, Canada; 3Clinical Nutrition and Risk Factor Modification Centre, St. Michael’s Hospital, Toronto, ON M5B 1W8, Canada; 4Toronto 3D Knowledge Synthesis and Clinical Trials Unit, Toronto, ON M5B 1W8, Canada; 5Li Ka Shing Knowledge Institute, St. Michael’s Hospital, Toronto, ON M5B 1W8, Canada; 6Division of Endocrinology and Metabolism, St. Michael’s Hospital, Toronto, ON M5B 1W8, Canada

**Keywords:** total sugars, added sugars, free sugars, consumption, Canadian, adults, dietary assessment, cross-sectional, macronutrients, micronutrients, food categories

## Abstract

**Background:** Global dietary guidelines recommend reducing free sugars intake, which may affect choices of sugars-containing foods, including important sources of key micronutrients. The purpose of the study was to compare the intakes of nutrients stratified by intakes of sugars in Canadian adults. **Methods:** The first-day 24-h dietary recalls from adults (*n* = 11,817) in the 2015 Canadian Community Health Survey-Nutrition were used to compare macronutrients, micronutrients and food categories across quintiles of total sugars [by %energy (%E)], adjusted for misreporting status and covariates. **Results:** Canadian adults consumed on average 86.9 g/day (18.8 %E) from total sugars and 47.5 g/day (9.9 %E) from free sugars. Mean intakes for the 1st (Q1), 3rd (Q3) and 5th (Q5) quintiles of total sugars were 7.9%E, 18.3%E and 33.3%E, respectively. Q3 had higher fibre, calcium, vitamin D, vitamin A, vitamin C and potassium intakes than Q1 (*p* < 0.001), reflecting higher fruit, milk and yogurt (*p* < 0.001) consumption. Compared to Q5, Q3 had higher intakes of folate, vitamin B_12_, iron and zinc. **Conclusion:** This study provides the first detailed analyses of Canadian adults’ macro- and micro-nutrient intakes stratified by different intakes of total sugars. Moderate intakes of total sugars may result in greater intakes of fibre and micronutrients. Overall nutrient intake should be considered when making food choices.

## 1. Introduction

Global concerns regarding the prevalence of obesity and excess energy intake have sparked increasing interest in understanding potential associations with specific dietary components, among which sugars have been a focus of research and media attention. Systematic reviews and meta-analyses of randomized controlled trials suggest that sugars do not have a causal effect on obesity, diabetes or other metabolic diseases beyond their contribution to total energy intake [1,2,3,4]. Sugars-sweetened beverages specifically, on the other hand, have been shown to be the major source of sugars in the diet that adversely affect risk factors of cardiometabolic outcomes in prospective cohort studies [5,6,7,8,9,10,11], as well as in randomized controlled trials when consumed as a source of excess calories [12,13,14,15,16,17,18,19,20]. Conversely, systematic reviews and meta-analyses have demonstrated that other sugars-containing foods, such as fruits and yogurts, have not shown negative health impacts on risk of certain chronic diseases [21,22] or major cardiometabolic risk factors [23]. 

Nevertheless, public health agencies such as the World Health Organization (WHO), recommend limiting free sugars intake to less than 10% of energy (%E) based on moderate quality evidence related to dental caries [24]. A similar quantitative recommendation of less than 10%E has been published in the US Dietary Guidelines for Americans 2015–2020, largely based on food pattern modelling studies and national data showing that people with free sugars intake above this level may not be able to meet food group and nutrient needs without over-consuming dietary energy [25]. Other jurisdictions, such as the United Kingdom, France and Nordic countries, have also published similar quantitative guidelines for added or free sugars [26,27,28]. While the scientific bases of various sugars guidelines are still frequently discussed [29,30,31], these recommendations often take a nutrient-based rather than a food-based approach. 

Worldwide sugars consumption has been either stable or decreasing in both absolute (gram per day) and relative (%E) terms, based on 44 comparisons within 13 countries [32]. More recent dietary survey data from the United States, Australia, the United Kingdom, Germany and the Netherlands also confirmed these observations [33,34,35,36,37,38]. Using data from the two most recent iterations of the Canadian Community Health Survey (CCHS) (2015 CCHS-Nutrition and 2004 CCHS-Nutrition), Statistics Canada estimated that among plausible reporters, Canadians consumed on average less total sugars in 2015 compared to 2004, which was largely driven by a decrease in free sugars coming from soft drinks [39]. This trend mirrors the data from recent consumer surveys in Canada showing that about 50% of Canadians made an effort to consume less sugars [40]. However, it remains unclear whether altered food choices based solely on one nutrient, sugars, may result in unintentional changes to the intakes of certain sugars-containing foods that are nutrient-dense and good sources of fibre, calcium and vitamin D, thus affecting overall nutrient intake profiles. Therefore, the objective of this study was to assess the consumption of sugars (total, added and free) in Canadian adults and compare the intakes of macro- and micronutrients, as well as food categories, across the spectrum of sugars intakes, using the 2015 CCHS-Nutrition Public Use Microdata Files (PUMF).

## 2. Materials and Methods

### 2.1. Data Source

The 2015 CCHS-Nutrition provides the most up-to-date nutrition data available for Canadians. The details of this cross-sectional, voluntary, nationally representative health survey conducted by Statistics Canada have been published elsewhere [41]. The United States Department of Agriculture (USDA) five-step automated multi-pass method (AMPM), adapted and modified for the Canadian population, was used to conduct computer-assisted interviews [42]. Survey interviews were conducted between January 2, 2015 and December 31, 2015 and consisted of two separate questionnaires per household: 1) 24-hour dietary recall to assess all food and beverage intake of an individual over 24 hours, and 2) a general health questionnaire to collect sociodemographic, anthropometric and health status data. In the 2015 CCHS—Nutrition, 20,487 people completed the initial 24-hour dietary recall and a random subsample of 7608 completed the second recall by phone 3–10 days after; response rates were 61.6% and 68.6%, respectively. For the present analysis, the first 24-hour dietary recall from the 2015 CCHS-Nutrition PUMF data was used [43]. This is the first PUMF released for CCHS-Nutrition surveys that the general public can readily access for free. It contains all the information required for the purpose of this study, such as demographic, dietary (including details on what foods were eaten and quantities), and health information (including self-reported physical activity, height and weight, health conditions) [43].

### 2.2. Participants and Data Collection

The target population for 2015 CCHS-Nutrition includes all individuals aged 1 year and above living in private dwellings in the 10 Canadian provinces, and did not include individuals who were full-time members of the Canadian Forces or who lived in the Territories, on reserves and other Aboriginal settlements, in some remote areas, or in institutions (e.g., prisons or care facilities) [44]. A complete description of the sampling method used for the 2015 CCHS-Nutrition is provided elsewhere [41]. In brief, the sampling frame was created from Canada’s Census Dissemination Areas. Samples were selected to represent the population in terms of age, sex, geography and socioeconomic status through a three-stage sampling within the frame. Seasonality was taken into account, which means there was an even distribution of the sample throughout the year, as well as an even representation of rural and urban areas throughout the year [41]. For the present analysis, participants were removed if they were <19 years of age, breastfeeding, or if they had invalid dietary recalls [45]. The 2015 CCHS-Nutrition PUMF does not identify pregnant participants. The final sample included 11,817 adult (≥ 19 years; males and females) respondents (Figure 1). Foods and beverages reported in the 24-hour dietary recall were coded by Health Canada using the 2015 Canadian Nutrient File and Health Canada Bureau of Nutritional Science (BNS) food codes [44,46].

### 2.3. Estimation of Added and Free Sugars

For our analyses, added and free sugars intake variables were created for each person based on total sugars using the 10-step decision algorithm previously published by Louie et al [48]. Added sugars were defined as sugars and syrups that are added to foods by manufacturers or consumers. Free sugars were defined according to the WHO definition as sugars added to foods by manufacturers or consumers, as well as those naturally present in honey, syrups and fruit juices; representing added sugars as well as sugars in 100% fruit juices. Of the 177 BNS food codes, most of the values of added sugars and free sugars variables were assigned based on objective measures (steps 1–6 in [48]) and about 6% were assigned subjectively (steps 7–10 in [48]). With respect to individual food categories per BNS code, it was assumed that all sugars in fruits, fruit juices, vegetables, and unsweetened milk were naturally-occurring, while all sugars in confectionary, sugars and syrups, fruit drinks, baked goods and products, frozen desserts were considered added. The amounts of added sugars in foods with both naturally-occurring and added sugars such as sweetened yogurt were determined by assessing the difference of their total sugars content in comparison to their unsweetened counterparts. 

### 2.4. Adjusting for Misreporting

Based on the method proposed by Garriguet [47], reported energy intakes (EI) were compared to each respondent’s total energy expenditure (TEE) to identify misreporters (i.e., over- and under-reporting of energy intake). Briefly, respondents’ TEE were predicted based on the variables age, sex, height, weight and physical activity levels using the Institute of Medicine equations [49]. Body Mass Index (BMI) was calculated using measured height and weight for the sample of respondents who had these measurements whereas a correction factor was applied to the adults with self-reported height and weight. Cut-offs to define sedentary, low active, active and very active for physical activity levels were applied from Health Canada’s Reference Guide to Understanding and Using the Data from 2015 CCHS-Nutrition [44]. Underweight respondents (19 years of age and older with a BMI < 18.5 kg/m^2^, *n* = 250) were excluded since there was no TEE equation for underweight subjects to identify misreporting. Under- and over-reporting were defined as the ratio of EI:TEE < 0.7 and >1.42, respectively, whereas those in between were considered plausible reporters. The resulting misreporting variable (“misreporting status”) was used as a covariate in all analyses. 

### 2.5. Statistical Analysis

Analyses were completed using SAS version 9.4 (SAS Institute). In order to ensure national representation, estimates were weighted using the survey weights provided by Statistics Canada. Variance estimation was performed using the bootstrap balanced repeated (BRR) replication with 500 replicates. All nutrients were expressed as either a % of energy (%E) or per 1000 kcal to adjust for energy. To satisfy the normality requirement of statistical tests, the values of the nutrients were transformed to approximate normal distribution using the Box-Cox method [50,51]. Food category variables were determined based on Health Canada BNS groupings and expressed as weight (grams/day) (Appendix A). 

Intakes of total, free and added sugars in Canadian adults were reported and comparisons were made across different age groups for each sex, and between males and females within each age group. The sample was divided into evenly distributed quintiles based on intakes of total sugars as %E, to assess and compare the intakes of macronutrients, and micronutrients, as well as food categories across quintiles. Comparisons were also made within different age groups for each sex across quintiles. Covariate-adjusted comparisons between quintiles of total sugars intake for macronutrients, micronutrients and food categories were determined using the ANOVA method with post-hoc Bonferroni adjustment for multiple comparisons (PROC SURVEYREG). Three covariate-adjusted models were completed. For analyses of food categories, Model 1 (which adjusted for “misreporting status”), and Model 2 (which adjusted for “misreporting status”, and also included the following covariates: energy intake, age, sex, smoking, self-perceived health, blood pressure, diabetes, heart disease, cancer, osteoporosis, education, physical activity, income, BMI, immigrant status and weekend reference day) were used. For analyses of macronutrients and micronutrients, Model 1 and Model 3 (which included all covariates in Model 2 except energy intake, since the data on macronutrients and macronutrients were already adjusted for energy intake as either %E or per 1000 kcal) were used. Covariates were selected from the literature and were also examined in a backward selection regression model. Results from the most-adjusted model with two-tailed *p*-value < 0.05 were reported as statistically significant. 

## 3. Results

### 3.1. Sample Characteristics

The sample evenly represented males (50.6%) and females (49.4%), with a mean age of 49.4 [standard error (SE) 0.3] years and a mean BMI of 27.7(SE 0.1) kg/m^2^. About 44% of Canadian adults met the Canadian Physical Activity Guideline of greater than 150 minutes of moderate to vigorous aerobic physical activity per week, with a mean of 3.5 (SE 0.1) hours/week for the entire sample. The frequency distributions of key demographic information of the study sample used in these analyses are described in Appendix A. The consumption of total sugars as %E was associated with sex (*p* < 0.0001), age/sex groupings (*p* = 0.012), self-perceived health (*p* = 0.034) and diabetes (*p* = 0.0484), but not with type of smoker, self-perceived health, BMI, education, income, or whether or not the subjects had high blood pressure, heart disease, cancer or osteoporosis. 

### 3.2. Consumption of Total, Free and Added Sugars by Age and Sex Groups

In 2015, the mean consumption of total sugars for Canadian adults was 86.9 (SE 1.0) g/day, equivalent to 18.8 (SE 0.2) %E (Figure 2). Slightly less than half of the total sugars came from added sugars at 41.7 (SE 0.8) g/day, accounting for 8.6 (SE 0.1) %E. The consumption of free sugars was 47.5 (SE 0.9) g/day or 9.9 (SE 0.1) %E (Figure 2). When comparing males and females within the same age range, males in most age groups had higher intakes of absolute amounts of total, free and added sugars (as g/day) compared to females (Figure 2A). However, there were no significant differences between males and females when total, added or free sugars intakes were expressed as %E, except for the age group 50–70 years where males had a significantly lower %E from total sugars compared to females (Figure 2B). When the same sex was compared across different age groups, males of 19–30 years had lower %E from free sugars compared to males of 30–50 years and 50–70 years. 

### 3.3. Consumption of Major Sugars-Containing Foods Categories

The mean intakes of total sugars as %E for Q1, Q3 and Q5 were 7.9 (SE 0.1) %E, 18.3 (SE 0.01) %E and 33.3 (SE 0.3) %E, respectively. These quintiles were utilized for further analyses of food categories and nutrient intakes. Compared to Canadian adults with low intakes of total sugars (Q1), the intakes of cakes, pies, danishes and other pastries, frozen dairy products, milks, fruit, sugars, syrups, preserves, confectionary, fruit juice and regular soft drinks were significantly higher in those with both moderate (Q3) and high (Q5) intakes of total sugars (*p* < 0.05, Table 1). In addition, Q5 also had higher intakes of pasta, rice, cereal grains and flour, fruit, sugars, syrups, preserves, confectionary, fruit juice, regular soft drinks and fruit drinks compared to Q3 (*p* < 0.05, Table 1). Interestingly, there was no difference observed between quintiles in the intakes of diet soft drinks, or important sources of carbohydrates such as white breads, wholemeal breads, whole grain and high-fibre breakfast cereals, vegetables, vegetable soups and legumes (Table 1, Appendix A). 

### 3.4. Consumption of Macronutrients and Micronutrients 

Total sugars accounted for 20%, 38.3% and 56.5% of total carbohydrates in Q1, Q3 and Q5, respectively (Table 2 and Appendix A). The %E from carbohydrates, total, natural, added and free sugars were significantly different across quintiles (*p* < 0.0001). Canadian adults with the lowest intake of total sugars (%E) (i.e., Q1) had a significantly higher %E from total fat and all subcategories of dietary fat except alpha-linolenic acid compared to Q3 and Q5 (*p* < 0.05). In addition, Q1 had the lowest fibre intake (g/1000 kcal) and the highest intake of alcohol compared to Q3 and Q5 (*p* < 0.05). Q5 had the lowest total energy compared to Q3 (*p* < 0.05). With regards to micronutrients, Q3 had higher vitamin A, riboflavin, vitamin C, vitamin D, calcium and potassium compared to Q1, and higher folate, vitamin B_12_, iron, phosphorus and zinc compared to Q5 (*p* < 0.05 for each). The sodium intake in Q3 was significantly higher than in Q5 but lower than Q1 (*p* < 0.05). Overall, Q3 with moderate intakes of total sugars had the highest intakes of 11 micronutrients among the 16 micronutrients reported in 2015 CCHS—Nutrition, and for 13 micronutrients Q3 had statistically greater intakes compared to the quintile with the lowest intake. Nutrient intakes across quintiles within different sex and age groups followed similar trends (Appendix A).

## 4. Discussion 

### 4.1. Canadian Consumption of Sugars

Canadian adults on average consumed 87 g/day of total sugars in 2015, which is similar to that reported recently by Statistics Canada using 2015 CCHS-Nutrition data [39]. In the Statistics Canada report, there was no information on the intakes of added or free sugars as %E, which are most commonly the focus of global dietary guidelines [24,26]. The present analysis demonstrated that Canadian adults consumed 18.8%E (87 g/day) from total sugars, with less than half coming from added sugars at 8.6%E (42 g/day) and 9.9%E (48 g/day) coming from free sugars, with no differences between men and women (as %E). In the previous CCHS conducted in 2004, total sugars intake represented a daily average of 20.0%E for adults, where 9.9%E came from added sugars and 11.4%E came from free sugars based on crude estimation [52]. Although it appears sugars consumption has declined over the past decade, which follows global consumption trends [32,33,34,37] and consumer surveys [40,53], it should be noted that differences in population demographics (e.g., older age), survey methods and under-reporting status may have influenced these observations [47].

Compared to a contribution of 11.7 %E from free sugars in a cohort of French-speaking adults living in Quebec Canada (the PREDISE Study) [54], this analysis of 2015 CCHS-Nutrition showed a lower national average for free sugars (9.9%E), which may in part be related to a slightly different age range (19 years and older in 2015 CCHS-Nutrition as compared to 18 to 65 years old in the PREDISE Study), as well as some unique cultural traditions, lifestyle and dietary habits and food choices among Quebec residents [55,56]. Compared to sugars intake in other developed countries, added sugars intake in Canadian adults (8.6%E, 42 g/day) was about 30% lower than recent reports for American adults (13.1%E for added sugars, NHANES 2009–2012) and Australian adults (60.3 g/day for added sugars, Australian Health Survey 2011–2012) [36,57]. Canadian intake of free sugars (9.9%E, 48 g/day) was about 10% lower than adults in the United Kingdom (11.1%E, the National Diet and Nutrition Survey 2014–2016), Switzerland (11%E, Swiss National Nutrition Survey 2014–2015) and New Zealand (11.1%E, New Zealand Adult Nutrition Survey 2008–2009), but higher than Spain (7.1%E, the Anthropometry, Intake and Energy Balance Study 2013) and Japan (6.1%E for men and 7.4%E for women in 2013) [37,58,59,60,61]. Although there is no specific quantitative dietary recommendation for sugars in Canada, the average adult intakes of both free and added sugars are very close to the 2015 WHO 10%E free sugars guideline and Dietary Guidance for Americans 2015–2020 of limiting added sugars intake to less than 10%E [24,25]. 

Amongst the quintiles of total sugars intake compared in this study, the mean free sugars intake of Q3 was 9.2%E, similar to the population mean of 9.9 %E and just below the 10 %E threshold recommended by the WHO [24]. The mean free sugars intake of Q1 at 3.2 %E is less than the conditional recommendation of 5%E by the WHO (based on very low-quality evidence related to dental caries) and by the UK Scientific Advisory Committee on Nutrition [26]. At the high end of intakes, Q5 had a mean intake of 20.1%E from free sugars, twice the recommended intake by the WHO. Nevertheless, meeting such single nutrient-based dietary guidelines may not always correspond to improved overall diet quality. Thus, nutrient intake data in Q1, Q3 and Q5 allow for a comparison of groups consuming low, moderate and high intakes of sugars.

### 4.2. Food Categories

Compared to those with the lowest sugars intakes (Q1), Canadians with the highest sugars intake (Q5) consumed more sweetened baked products, sugars and confectionary, fruit drinks, and regular soft drinks, but also consumed more sources of naturally-occurring sugars including about three times as much fruit, milk and yogurts. On the other hand, it appears that Canadian adults with moderate intakes of total sugars (Q3) consumed moderate amounts of many sugars-containing foods and beverages such as fruit drinks, regular soft drinks, and sugars and confectionary, compared to the much higher amounts in Q5. Fruit and fruit juice intakes in Q3 were higher than Q1 but lower than Q5. The consumption of milk, frozen dairy, and cakes, pies and pastries for Q3 and Q5 were higher compared to Q1, but no difference was observed in these categories between Q3 and Q5. 

### 4.3. Macronutrient Intakes Across Different Total Sugars Intake Levels

The mean carbohydrate intake of Q1 (38.9%E) was below the accepted macronutrient distribution range (AMDR) of 45–65% for adults [49], and significantly lower than those of Q3 (47.7%E) and Q5 (58.4%E). On the other hand, those with a moderate intake of carbohydrate in Q3 had higher intakes of fibre than Q1 and had moderate intakes of free and added sugars that are close to dietary recommendations. While the current study was unable to directly associate nutrient intakes with the health status of the population, a previous meta-analysis of prospective cohort studies suggests that all-cause mortality is lowest when carbohydrate intake contributes about 50–55% of daily energy intake [62]. 

The mean fat intake in Q1 (36.1%E) was greater than the AMDR for fat (20–35%E for adults) [49] and significantly higher than those of Q3 (32.2%E) and Q5 (26.4%E). There was an inverse trend of intakes between total sugars and total fat intakes across quintiles; this mirrors the “sugar-fat seesaw” phenomenon [63]. This reciprocal relationship was also observed for saturated fat, where the mean intake of saturated fat in Q1 was 11.3%E, significantly greater than Q3 and Q5, and exceeding Dietary Reference Intake (DRI) recommendations to limit to <10%E [49]. Protein intake was significantly greater in Q1 compared to Q5 (18.8%E vs. 13.9%), although all levels were within the AMDR (10–35%). Dietary cholesterol intake was also significantly greater in Q1 (178.1 mg/1000 kcal) compared to Q5 (109.9 mg/1000 kcal), though guidelines no longer recommend an upper limit for dietary cholesterol. 

Dietary fibre intake was significantly lower in Q1 (16.7 g/day) compared to either Q3 (19.4 g/day) or Q5 (18.6 g/day), however each group mean was below the DRI recommendation of 25 g/day for women and 38 g/day for men [49]. Fibre is a nutrient of particular interest, since most Canadians only consume about half of the recommended amount [64] and there is strong evidence that dietary fibre reduces the risk of colon cancer, cardiovascular diseases and type 2 diabetes, and has been associated with a healthy microbiome [65,66,67,68,69]. Since there were no significant differences among groups in the intakes of common fibre sources, such as wholemeal breads, high fibre breakfast cereals, legumes, vegetables or vegetable soups, the significantly lower fibre intakes observed in Q1 may be attributed to a significantly lower fruit intake compared to other quintiles. 

Interestingly, there was a significant difference in total energy intake across quintiles where the lowest intake was in Q5 (1806 kcal vs. 1919 kcal in Q3 and 1954 kcal in Q1). Given the observational nature of this study, this finding must be interpreted with caution, as some studies show a positive association between intakes of sugars (or sugar containing beverages) and energy intake [26,70,71,72,73]. However, it certainly confirms the importance of investigating the dietary pattern of higher and lower consumers of sugars, whose dietary choices may limit energy intake from certain nutrient-dense food sources. 

### 4.4. Micronutrient Intakes Across Different Total Sugars Intake Levels

Micronutrients currently at risk of insufficient consumption among Canadian adults include calcium, potassium and vitamin D [74,75]. The present study demonstrates that calcium, potassium and vitamin D intakes were lowest in Canadians with the lowest sugars intake (Q1) but were similar in those with moderate (Q3) or high sugars intakes (Q5). The low intakes of calcium and vitamin D in Q1 may be explained by the corresponding low intakes of dairy products such as milk and yogurt, which are the most readily available dietary sources of calcium and vitamin D. Adequate calcium intake has many health benefits including prevention of osteoporosis and colorectal adenomas, lower cholesterol values, and a reduction in hypertensive disorders in pregnancy [76,77,78,79,80], whereas vitamin D has been shown to have important skeletal and non-skeletal effects, including on cardiovascular function, glucose homeostasis and immune function [81,82,83,84]. The mean intake of calcium in Canadian adults is below recommendations for most age and sex groups, and has shown a declining trend between 2004 and 2015 [85,86]. Similarly, the majority of Canadians do not meet the adequate intake level for potassium or vitamin D [74,75], and low vitamin D intake may further reduce calcium absorption, exacerbating the challenge of meeting calcium recommendations. Potassium intake has been shown to reduce blood pressure, decrease risk of cardiovascular disease, and antagonize the negative consequences of high sodium consumption [87]. The low potassium intake observed in Q1 compared to Q3 and Q5 may be explained by the much lower corresponding intake of fruits, which may also explain the lower intakes of niacin, vitamin A, riboflavin and vitamin C. Since the present analysis demonstrates that Canadians with the lowest sugars intakes also have the lowest intake of key micronutrients, it is important to understand whether the recommendations to lower sugars intakes (such as the WHO conditional recommendation of limiting free sugars to less than 5%E), when followed without a whole diet perspective, may potentially have unintended consequences for the intakes of other important nutrients. It is also important to be aware of the amount of sugars present in foods that are either naturally rich in or fortified with calcium and vitamin D (e.g., orange juice) when trying to increase the intake of these nutrients.

Furthermore, when considering all 16 micronutrient intakes assessed in the 2015 CCHS-Nutrition, Canadians with a moderate intake of total sugars (Q3) had the highest intakes of most micronutrients compared to all other quintiles. Specifically, Q3 had higher vitamin A, riboflavin, vitamin C, vitamin D, calcium and potassium intakes compared to Q1, and higher folate, vitamin B_12_, iron, phosphorus and zinc intakes compared to Q5. These data suggest that Canadian adults with moderate intakes of total sugars appear to have better overall nutrient intakes than those with high or low intakes of total sugars, which is supported by a systematic review on sugars and micronutrient adequacy [88]. Similar observations have also been documented in recent cohorts in Australia and the US [89,90,91]. 

### 4.5. Strengths and Limitations

This study provides the most recent snapshot of the consumption of added and free sugars (as %E) among Canadian adults, as well as the first report on their macro- and micro-nutrient intake portfolio across different levels of sugars intakes. It is also the first published study on sugars consumption using the publicly available CCHS PUMF database on Nutrition, which offers researchers a much easier, time-efficient and cost-effective way to conduct research in comparison to the traditional access through Statistics Canada’s Research Data Centres [92]. There are also some important limitations to be considered when interpreting findings from this study. Due to restrictions in sampling the population, the results are not generalizable to residents of the three territories, full-time members of the Canadian Forces, people living on reserves or other Aboriginal settlements, people living in institutions, and people with a BMI categorized as underweight, since there is no TEE equation for underweight subjects, which is needed to identify misreporting. Data from only one 24-hour dietary recall was analyzed, which does not represent the subjects’ usual intake or imply nutrient adequacy, and should be interpreted with caution when comparing to dietary recommendations. However, systematic error has been shown to be less substantial in 24-hour recalls compared to other dietary assessments such as food frequency questionnaires [93]. The assessment of most variables used in the model were self-reported, which may have limited accuracy. Although mis-reporting status was adjusted as a covariate in our analysis, interpretation of self-reported dietary intake data should be done with caution, as in most studies involving dietary data, energy intake tends to be underestimated [47]; thus, the absolute amounts of sugars and other nutrients may have been underestimated compared to true intake levels. To overcome this inherent limitation, all nutrients were analyzed with energy adjustment so as to minimize the impact of under-reporting. 

## 5. Conclusions

In 2015, Canadian adults consumed on average 18.8 %E from total sugars and 9.9 %E and 8.6 %E from free and added sugars, respectively. A moderate intake of sugars may result in greater intakes of dietary fibre and key micronutrients such as calcium, vitamin D and potassium compared to high and low intakes of sugars. Dietary recommendations are shifting away from focusing on single nutrients and more on dietary patterns. However, current guidelines still recommend limiting sugars intake, which may lead to alterations in food choices, such as limiting fruit, milk and yogurt, which contain sugars but are also nutrient-dense. This may have unintended consequences for intakes of other important nutrients vital to good health including dietary fibre, potassium, calcium and vitamin D, which the majority of Canadian adults have low intakes of compared to dietary recommendations. Thus, the overall nutrient intake profile should be considered when making food choices. These findings also underscore the need for future research to understand the nutrient adequacy of Canadians with different intakes of sugars and prospectively monitor their diet quality when intentionally making changes to their sugars intake.

## Figures and Tables

**Figure 1 nutrients-12-01124-f001:**
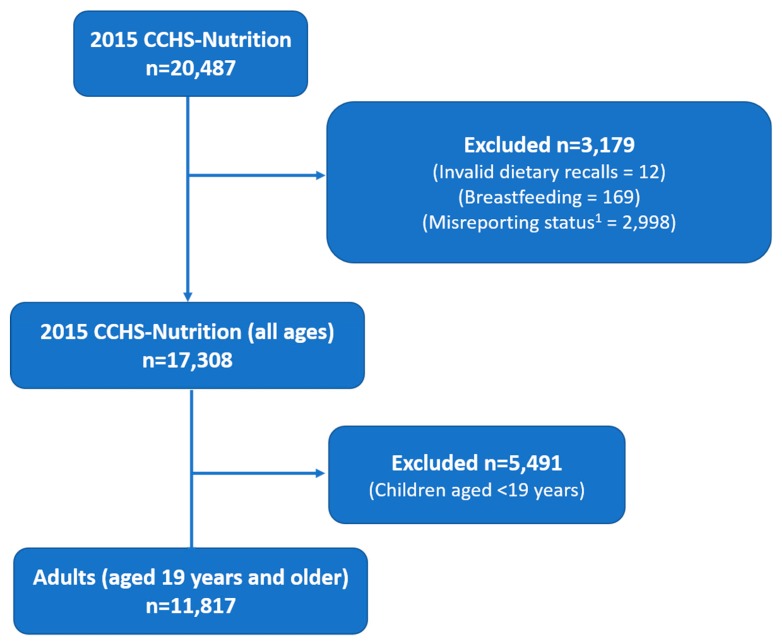
Sample selection process. ^1^ Did not have relevant information to adjust for misreporting status based on previously published method [47].

**Figure 2 nutrients-12-01124-f002:**
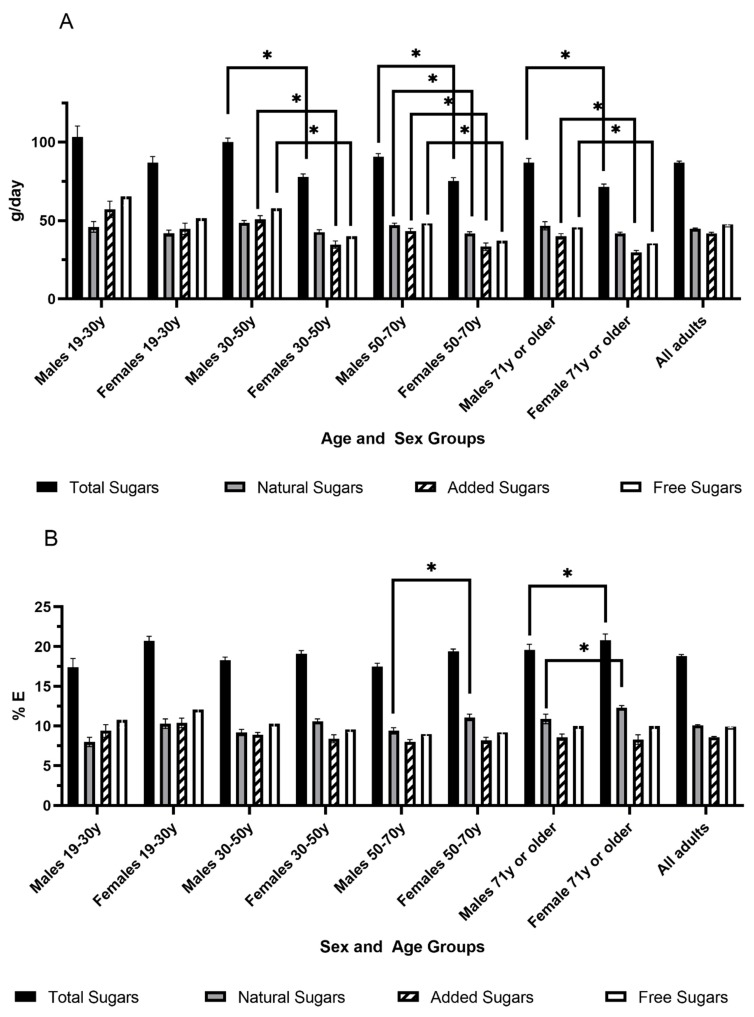
Intake of total, natural, free and added sugars as g/day (A) and % energy/day (B) in Canadian adults (*n* = 11,817, aged 19 years and older) by age and sex groups. Values are expressed as Mean and Standard Error (SE). y: years old. * *p* < 0.05 compared to females of the same age range using a general linear model (PROC SURVEYREG), including the following covariates (Model 3): misreporting status (determined using previously published methods by Garriguet [47]), age, sex, smoking, self-perceived health, blood pressure, diabetes, heart disease, cancer, osteoporosis, education, physical activity, income, BMI, immigrant status and weekend reference day. The n number in each age group is: males 19–30 y: 765, females 19–30 y: 757, males 31–50 y: 1839, females 31–50 y: 1945, males 51–70 y: 1961, females 51–70 y: 2105, males 71 y+: 1105, females 71 y+: 1340.

**Table 1 nutrients-12-01124-t001:** Adjusted means (g/day) and confidence intervals of major sugars-containing food categories and comparisons across quintiles of total sugars intake (% energy) in all adults (*n* = 11,817) in Models 1 and 2 *.

	Q1				Q3				Q5				Model 1	Model 2
	*N* = 2364				*N* = 2363				*N* = 2364					
	Model 1		Model 2		Model 1		Model 2		Model 1		Model 2			
Food Category	Mean	95% CI	Mean	95% CI	Mean	95% CI	Mean	95% CI	Mean	95% CI	Mean	95% CI	*p*-Value **	*p*-Value **
Pasta, rice, cereal grains and flour	112.3	83.6, 141.0	100.4 ^¥^	75.5, 125.2	83.9	71.7, 96.2	74.3 ^¥^	64.0, 84.7	53.6	41.5, 65.8	49.4	39.3, 59.4	<0.0001	<0.0001
White breads	21.0	17.7, 24.3	20.6	15.6, 25.6	31.2	22.5, 39.9	32.1	26.6, 37.7	22.9	19.6, 26.3	24.3	21.4, 27.3	0.1833	0.067
Wholemeal breads	21.6	18.0, 25.2	23.5	20.6, 26.4	21.7	15.9, 27.6	22.9	18.1, 27.7	16.6	12.8, 20.5	18.2	14.6, 21.8	0.0519	0.114
Other breads	43.9	36.6, 51.1	40.4 ^¥^	34.5, 46.2	35.4	25.8, 45.1	32.6	24.5, 40.8	26.7	21.6, 31.9	26.8	22.9, 30.8	<0.0001	0.001
Wholegrain and high fibre breakfast cereals	15.0	8.0, 22.0	16.0	8.7, 23.3	19.7	13.6, 25.9	19.7	13.6, 25.8	18.6	15.2, 21.9	19.9	16.4, 23.3	0.6212	0.787
Other breakfast cereals	1.1	0.1, 2.1	1.1 ^¥^	0.2, 2.1	1.9	0.7, 3.1	2.0	0.7, 3.3	2.8	1.6, 4.0	3.0	1.8, 4.2	0.0068	0.004
Cookies, biscuits and granola bars	7.0	4.7, 9.3	7.1 ^¥^	5.3, 8.8	10.1	8.2, 12.0	10.2	7.8, 12.5	11.2	9.3, 13.2	11.7	9.9, 13.6	<0.0001	<0.0001
Cakes, pies, danishes and other pastries	13.3	9.7, 16.9	14.4 ^¥,£^	8.7, 20.0	30.4	25.2, 35.7	30.9	26.0, 35.8	35.7	28.6, 42.8	38.3	31.1, 45.4	<0.0001	<0.0001
Frozen dairy products	2.3	1.1, 3.5	1.8 ^¥,£^	0.6, 3.1	16.1	8.0, 24.2	15.8	9.3, 22.4	22.6	17.7, 27.5	23.2	17.9, 28.6	<0.0001	<0.0001
Milks	58.7	49.6, 67.9	56.4 ^¥,£^	47.0, 65.9	170.7	154.2, 187.2	169.7	154.0, 185.3	190.5	161.4, 219.5	198.4	170.5, 226.3	<0.0001	<0.0001
Yogurts (natural and with fruits)	11.4	7.0, 15.7	12.7 ^¥^	9.0, 16.4	26.8	18.9, 34.7	26.3	17.0, 35.6	33.3	26.9, 39.6	33.4	27.5, 39.4	<0.0001	<0.0001
Fruit	59.7	51.9, 67.5	56.1 ^¥,£^	48.3, 63.9	160.4	145.0, 175.8	152.7 ^¥^	139.6, 165.8	223.6	202.4, 244.7	224.2	203.8, 244.6	<0.0001	<0.0001
Sugars, syrups, preserves	4.4	3.6, 5.1	3.8 ^¥,£^	2.6, 4.9	10.7	9.2, 12.2	10.5 ^¥^	9.2, 11.7	20.4	16.8, 24.1	20.7	17.2, 24.2	<0.0001	<0.0001
Confectionary	2.5	1.7, 3.3	2.5 ^¥,£^	1.3, 3.6	9.1	6.0, 12.2	8.9 ^¥^	6.1, 11.7	24.2	17.5, 30.9	24.7	18.0, 31.3	<0.0001	<0.0001
Fruit juice	23.4	15.6, 31.3	17.4	10.1, 24.7	58.9	51.1, 66.8	54.8 ^¥^	47.3, 62.3	112.6	96.6, 128.5	112.2	96.8, 127.6	<0.0001	<0.0001
Alcoholic beverages	326.8	244.2, 409.5	311.0 ^¥,£^	233.6, 388.4	116.2	79.5, 152.9	114.5 ^¥^	96.1, 132.9	52.5	34.9, 70.1	59.5	42.9, 76.1	<0.0001	<0.0001
Tea, coffee	1511.1	1441.0, 1581.2	1482.4 ^¥^	1414.5, 1550.2	1555.6	1460.9, 1650.4	1523.1 ^¥^	1451.0, 1595.2	1308.7	1240.5, 1377.0	1305.1	1243.4, 1366.8	<0.0001	<0.0001
Soft drinks, regular	24.5	15.6, 33.4	12.1 ^¥,£^	0.7, 25.0	60.6	48.5, 72.6	58.5 ^¥^	41.4, 75.7	190.4	139.3, 241.5	191.9	144.4, 239.4	<0.0001	<0.0001
Soft drinks, diet	55.4	38.2, 72.6	55.7	40.9, 70.6	40.6	18.1, 63.0	43.5	23.4, 63.6	21.8	11.1, 32.5	26.3	15.3, 37.3	0.0109	0.014
Fruit drinks	11.1	4.1, 18.2	9.8 ^¥^	3.5, 16.1	18.2	10.0, 26.3	17.9 ^¥^	9.6, 26.1	53.5	39.9, 67.0	54.4	40.0, 68.8	<0.0001	<0.0001
Other beverages	3.4	0.8, 6.0	0.2	5.5, 5.8	9.5	0.1, 19.0	8.9	0.2, 18.0	27.1	3.5, 57.6	27.0	0.2, 53.7	0.0065	0.014
Soups	39.3	32.3, 46.3	41.5	34.2, 48.7	39.6	26.5, 52.7	41.3	27.7, 54.9	41.7	30.8, 52.6	43.1	33.4, 52.8	0.0571	0.088

* Intake of food categories across quintiles of total sugars intake (as % energy) were compared using a general linear model (PROC SURVEYREG) including the following covariates: Model 1-misreporting status (determined using previously published methods by Garriguet [47]); Model 2-misreporting status plus age, sex, smoking, self-perceived health, blood pressure, diabetes, heart disease, cancer, osteoporosis, education, physical activity, income, BMI, immigrant status and weekend reference day and total energy intake. Food categories are aligned with food categories presented in the 2015 Canadian Community Health Survey (CCHS)-Nutrition with few modifications (see Appendix A) and expressed as grams by weight. Note: Quintiles 2 and 4 are not presented for readability. Adjusted means and confidence intervals of non-essential food categories by quintiles of total sugars intake (as % energy) of Models 1 and 2 are presented in the Appendix A. Means and standard errors of all food categories across of total sugars intake (as % energy) in Models 1 and 2 are presented in the Appendix A. ** The *p*-value represents the global *p*-value for a significant difference across quintiles of intake of total sugars (% energy) **^¥^** significant difference compared to Q5 in post-hoc comparison with Bonferroni adjustment in Model 2 (*p* < 0.05) **^£^** significant difference compared to Q3 in post-hoc comparison with Bonferroni adjustment in Model 2 (*p* < 0.05). CI, confidence interval.

**Table 2 nutrients-12-01124-t002:** Means and 95% confidence intervals of intakes from macro- and micro-nutrients, and comparisons across quintiles of total sugars intake (% energy) in all adults (*n* = 11,817) in Models 1 and 3 *.

	Q1				Q3				Q5				Model 1	Model 3
	*N* = 2364				*N* = 2363				*N* = 2364					
	Model 1		Model 3		Model 1		Model 3		Model 1		Model 3			
	Mean	95% CI	Mean	95% CI	Mean	95% CI	Mean	95% CI	Mean	95% CI	Mean	95% CI	*p*-value **	*p*-value **
Energy (kcal)	1953.8	1903.8, 2003.8	1854.6	1819.1,1890.1	1919.2	1819.9, 2018.5	1893.7 ^¥^	1828.8,1958.5	1805.5	1763.9, 1847.2	1823.1	1789.6, 1856.6	<0.0001	0.012
Carbohydrates (%)	38.9	37.2, 40.5	38.9 ^¥,£^	37.4, 40.4	47.7	46.0, 49.4	47.6 ^¥^	46.3, 49.0	58.4	57.6, 59.3	58.3	57.5, 59.2	<0.0001	<0.0001
Total Fibre (g/1000 kcal)	8.6	8.2, 8.9	8.9 ^¥,£^	8.5, 9.2	10.1	9.4, 10.9	10.1	9.6, 10.6	10.3	9.9, 10.8	10.3	9.9, 10.7	<0.0001	<0.0001
Total Sugars (%)	7.9	7.7, 8.1	7.8 ^¥,£^	7.7, 8.0	18.3	18.3, 18.4	18.4 ^¥^	18.2, 18.5	33.0	32.4, 33.7	33.0	32.5, 33.6	<0.0001	<0.0001
Natural Sugars (%)	4.8	4.6, 5.0	5.0 ^¥,£^	4.8, 5.2	10.2	9.7, 10.7	10.1 ^¥^	9.8, 10.5	15.6	15.0, 16.3	15.7	15.0, 16.3	<0.0001	<0.0001
Added Sugars (%)	3.0	2.8, 3.1	2.8 ^¥,£^	2.6, 3.0	8.1	7.6, 8.6	8.2 ^¥^	7.8, 8.5	17.3	16.5, 18.1	17.3	16.5, 18.0	<0.0001	<0.0001
Free Sugars (%)	3.2	3.1, 3.4	3.0 ^¥,£^	2.8, 3.2	9.2	8.7, 9.7	9.2 ^¥^	8.8, 9.6	20.1	19.2, 20.9	20.0	19.2, 20.8	<0.0001	<0.0001
Fat (%)	36.1	35.1, 37.2	36.2 ^¥,£^	35.3, 37.1	32.2	31.1, 33.2	32.3 ^¥^	31.5, 33.1	26.4	25.7, 27.1	26.4	25.8, 27.0	<0.0001	<0.0001
Saturated Fat (%)	11.3	10.8, 11.7	11.4 ^¥,£^	11.0, 11.7	10.4	9.9, 11.0	10.5 ^¥^	10.1, 10.9	9.0	8.8, 9.3	9.1	8.8, 9.4	<0.0001	<0.0001
Monounsaturated Fat (%)	13.8	13.4, 14.3	13.8 ^¥,£^	13.4, 14.2	12.0	11.5, 12.5	12.0 ^¥^	11.6, 12.4	9.4	9.0, 9.8	9.4	9.0, 9.8	<0.0001	<0.0001
Polyunsaturated Fat (%)	7.8	7.4, 8.1	7.9 ^¥,£^	7.5, 8.2	6.8	6.6, 7.0	6.9 ^¥^	6.7, 7.1	5.5	5.2, 5.7	5.4	5.2, 5.7	<0.0001	<0.0001
Linoleic acid (%)	6.6	6.3, 7.0	6.7 ^¥,£^	6.4, 7.0	5.7	5.6, 5.9	5.8 ^¥^	5.6, 5.9	4.6	4.4, 4.9	4.6	4.4, 4.9	<0.0001	<0.0001
Alpha, linolenic Acid (%)	0.8	0.8, 0.8	0.8 ^¥^	0.8, 0.9	0.8	0.7, 0.8	0.8 ^¥^	0.7, 0.8	0.6	0.6, 0.7	0.6	0.6, 0.7	<0.0001	<0.0001
Protein (%)	18.8	17.9, 19.7	18.7 ^¥^	18.0, 19.5	17.5	17.1, 17.9	17.4 ^¥^	16.9, 17.9	13.9	13.3, 14.5	14.0	13.5, 14.4	<0.0001	<0.0001
Cholesterol (mg/1000 kcal)	178.1	162.6, 193.6	175.9 ^¥^	164.0, 187.8	153.2	139.5, 167.0	151.5 ^¥^	137.0, 165.9	109.9	101.2, 118.7	109.9	101.7, 118.1	<0.0001	<0.0001
Alcohol (%)	6.2	4.8, 7.6	6.2 ^¥,£^	4.8, 7.5	2.6	2.0, 3.3	2.7 ^¥^	2.2, 3.2	1.3	0.7, 1.8	1.3	0.8, 1.8	<0.0001	<0.0001
**Micronutrients: Per 1000 kcal**														
Vitamin A (ug RAE)	338.9	298.5, 379.4	349.9 ^£^	311.0, 388.7	370.6	349.8, 391.5	371.0	346.6, 395.5	378.7	340.8, 416.7	376.8	340.6, 413.1	<0.0001	<0.0001
Thiamin (mg)	0.9	0.8, 0.9	0.9	0.8, 0.9	0.9	0.8, 0.9	0.9	0.8, 1.0	0.8	0.8, 0.9	0.8	0.8, 0.9	0.003	0.006
Riboflavin (mg)	1.0 ^£^	1.0, 1.0	1.0	1.0, 1.1	1.1	1.1, 1.1	1.1	1.1, 1.1	1.1	1.0, 1.1	1.1	1.0, 1.1	<0.0001	<0.0001
Niacin (mg)	23.7	22.6, 24.8	23.7 ^¥,£^	22.6, 24.7	21.7	20.9, 22.5	21.7 ^¥^	21.0, 22.4	18.0	17.4, 18.5	18.0	17.5, 18.5	<0.0001	<0.0001
Vitamin B,6 (mg)	0.9	0.9, 1.0	0.9	0.9, 1.0	1.0	0.9, 1.0	0.9	0.9, 1.0	0.9	0.9, 0.9	0.9	0.9, 0.9	0.070	0.019
Folate (ug DFE)	247.8 ^¥^	238.8,256.8	246.9	238.0, 255.8	242.8	228.1, 257.5	240.0 ^¥^	226.7, 253.4	221.9	210.8, 233.1	219.0	207.2, 230.9	<0.0001	<0.0001
Vitamin B,12 (mg)	2.3	2.1,2.5	2.3	2.0, 2.6	2.4	2.1, 2.6	2.4 ^¥^	2.1, 2.6	1.9	1.7, 2.0	1.9	1.7, 2.0	0.013	0.011
Vitamin C (mg)	35.9 ^¥,£^	32.3,39.5	36.4	33.0, 39.9	56.2	50.2, 62.2	55.0	50.2, 59.9	85.1	74.0, 96.2	84.5	73.9, 95.2	<0.0001	<0.0001
Vitamin D (ug)	2.3 ^¥,£^	1.9,2.8	2.5	2.1, 2.8	2.7	2.5, 2.9	2.8	2.6, 2.9	2.6	2.4, 2.8	2.6	2.5, 2.8	<0.0001	<0.0001
Calcium (mg)	370.8 ^¥,£^	347.9,393.8	380.1	360.0, 400.3	439.8	421.8, 457.9	442.6	423.9, 461.3	458.2	441.6, 474.9	459.2	442.3, 476.2	<0.0001	<0.0001
Iron (mg)	6.7 ^¥^	6.5,6.9	6.8	6.6, 7.0	6.9	6.7, 7.2	6.9 ^¥^	6.7, 7.2	6.3	6.1, 6.5	6.3	6.1, 6.5	<0.0001	<0.0001
Magnesium (mg)	167.3	161.2,173.4	170.0	165.2, 174.9	177.7	170.7, 184.6	176.8	170.4, 183.2	169.7	163.9, 175.1	169.9	164.6, 175.1	<0.0001	0.007
Phosphorus (mg)	696.2 ^¥^	671.7,720.6	706.4	682.0, 370.8	704.8	688.8, 720.8	707.4^¥^	692.9, 722.0	652.0	635.5, 668.5	655.2	639.2, 671.1	<0.0001	<0.0001
Potassium (mg)	1378.1 ^¥,£^	1336.1,1420.0	1409.6	1365.0,1454.2	1536.7	1495.6,1577.8	1541.3	1506.0, 1576.6	1611.8	1560.3, 1663.3	1619.2	1570.0, 1668.4	<0.0001	<0.0001
Zinc (mg)	6.0 ^¥^	5.8, 6.2	6.0	5.8, 6.2	5.7	5.5, 5.9	5.7 ^¥^	5.6, 5.9	4.8	4.6, 5.0	4.8	4.7, 5.0	<0.0001	<0.0001
Sodium (mg)	1621.9 ^¥,£^	1567.9,1675.8	1629.0	1580.6,1677.5	1512.0	1473.5,1550.6	1523.2 ^¥^	1486.1, 1560.4	1255.1	1215.3, 1294.3	1257.4	1216.8, 1298.0	<0.0001	<0.0001

* Intake of macro- and micro-nutrients across quintiles of total sugars intake (as % energy) were compared using a general linear model (PROC SURVEYREG) including the following covariates: Model 1—misreporting status (determined using previously published methods by Garriguet [47]); Model 3—misreporting status plus age, sex, smoking, self-perceived health, blood pressure, diabetes, heart disease, cancer, osteoporosis, education, physical activity, income, BMI, immigrant status and weekend reference day. Note: Quintiles 2 and 4 are not presented for readability. Means and standard error of the means were presented in the Appendix A. ** The p-value represents the global p-value for a significant difference across quintiles of intake of total sugars (%E) **^¥^** significant difference compared to Q5 in post-hoc comparison with Bonferroni adjustment in Model 3 (*p* < 0.05). **^£^** significant difference compared to Q3 in post-hoc comparison with Bonferroni adjustment in Model 3 (*p* < 0.05) CI, confidence interval; BMI, body mass index; RAE, retinoic acid equivalent; DFE, dietary folate equivalents.

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
