# Peer review of "Canadian Adults with Moderate Intakes of Total Sugars have Greater Intakes of Fibre and Key Micronutrients: Results from the Canadian Community Health Survey 2015 Public Use Microdata File"

_nutrients, 2020, doi:10.3390/nu12041124_

Round 1

Reviewer 1 Report

Manuscript presented by Ye (Flora) Wang et al. concerns a very important and at the same time complex topic of consumption of total, added and free sugars in adults in relation to intake of macro- and micronutrients, and selected food categories.

Thank you for allowing me to review your study.

The concept is interesting and the idea worth pursuing,  because high sugar intake is a subject of much scientific debate due to the suggested health implications and recent free sugar recommendations by the WHO.

Nevertheless, the Authors did not escape some ambiguities in the text

There are several areas that need to be addressed to help improve the paper.

Please find below several  comments:

Title

It is worth considering another wording for the title - in its current form it is not very communicative

Abstract

It is important to emphasize what novelty this manuscript brings.

Keywords

add more words: adults, sugars (total, added, free), consumption

Introduction:

The introduction can be improved by giving more background of this covering population from other countries, e.g. Europe; emphasize the effect of added sugar on health.

An overview of existing (current) research requires some improvement.

Materials and Methods.

I suggest you add more information to the selection of participants.

Specify the total sample size and subgroups size accurately. It should also be provided in the tables.

Maybe it's better to present it in the diagram - (Flow chart of sample collection), including the inclusion/exclusion criteria for the study.

I would also suggest

Results

I would also suggest considering the order in which you discuss the results because, in my opinion, tables 2 and 3 should be swapped - first product groups then nutrients.

The presentation of results in tables needs to be improved.

Table 1: "All ages 19y or older" -  maybe better total?

Table 2 and 3: "Total Sugars, mean (min-max)" - does not apply to macro- and micronutrients or Food Category ???

Overall the tables are difficult to read and lack a uniform mode of data presentation. They would benefit from a clearer explanation of which values were compared where significant differences were detected.

Perhaps a much better solution would be to include 3 models in the manuscript - described in the Materials and Methods (Statistical Analysis)

Discussion

Make sure all of the discussed incidences are properly discussed and explained.

In my opinion, the discussion should be more comprehensive, covering more countries and on guidelines to limit added sugar intake.

In addition, the study's strengths should be highlighted.

It should also be noted that an increased intake of foods such as fortified products, e.g in Ca, Wit. D often increases the consumption of added sugar.

I would also suggest adding more key references. It is worth paying attention to e.g.

Della Corte KW, Perrar I, Penczynski KJ, Schwingshackl L, Herder C, Buyken AE. Effect of Dietary Sugar Intake on Biomarkers of Subclinical Inflammation: A Systematic Review and Meta-Analysis of Intervention Studies. Nutrients. 2018 May 12;10(5). pii: E606. doi: 10.3390/nu10050606.

Kearns C, Schillinger D. Guidelines to Limit Added Sugar Intake. Ann Intern Med. 2017 Aug 1;167(3):220. doi: 10.7326/L17-0257.

Sluik D, van Lee L, Engelen AI, Feskens EJ. Total, Free, and Added Sugar Consumption and Adherence to Guidelines: The Dutch National Food Consumption Survey 2007-2010. Nutrients. 2016 Jan 28;8(2):70. doi: 10.3390/nu8020070.

Hu, F.B. Resolved: There is sufficient scientific evidence that decreasing sugar-sweetened beverage consumption will reduce the prevalence of obesity and obesity-related diseases. Obes. Rev. 2013, 14, 606–619

All manuscript requires adaptation to editorial guidelines.

Reviewer 2 Report

The authors have made a great article that only has small changes, in which there may be a small improvement.

From my point of view, the results are interesting, and indicate that the average values for carbohydrate and sugar consumption are the most appropriate for the population. It may be interesting to mention a manuscript, Sara B Seidelmann et al August 16, 2018, Lancet Public Health 2018; 3: e419–28. The meta-analysis carried out in the PURE and ARIC studies can be seen in figure 3 of the manuscript, and may perhaps improve the discussion of the results.

The other aspect that can be improved is the presentation of the results, which is shown in Tables 1, 2, and 3. I throw the following question to the authors, it would be possible to reduce the tables with only the results with significant differences?. Or even better, transform the most relevant data into figures? This would probably make the manuscript easier to read, especially since it is possible to keep the data in the supplementary material, especially because there aren´t any figure in the manuscript, only tables.

For all this, I want to congratulate the authors for the fantastic manuscript they have written, and for taking the suggested changes into account.

Round 2

Reviewer 1 Report

This manuscript has been greatly approved. I can now support its application.
I just suggest changing the title of Figure 1. Maybe just: Sample collection chart.
However, I leave the final decision to the authors.

Best regards

Author Response

This manuscript has been greatly approved. I can now support its application. 
I just suggest changing the title of Figure 1. Maybe just: Sample collection chart. 
However, I leave the final decision to the authors.

Response: We thank the reviewer for the comments and appreciate your support of our work. The title of Figure 1 has been updated to "Sample collection chart" as suggested.